# Fire Impact on Carbon Pools and Basic Properties of Retisols in Native Spruce Forests of the European North and Central Siberia of Russia

Viktor V. Startsev *, Evgenia V. Yakovleva, Ivan N. Kutyavin and Alexey A. Dymov

Institute of Biology of Komi Science Centre of the Ural Branch, Russian Academy of Sciences, 28 Kommunisticheskaya St., 167982 Syktyvkar, Russia; kaleeva@ib.komisc.ru (E.V.Y.); kutjavin-ivan@rambler.ru (I.N.K.); aadymov@gmail.com (A.A.D.)
* Correspondence: vik.startsev@gmail.com; Tel.: +7-904-227-90-25

**Abstract:** Fires play an important role in the modern dynamics of boreal ecosystems. The article presents the results of studying the effect of old fires on soils and soil organic matter (SOM) of native spruce forests that were last affected by fires in the previous 100 to 200 years. The studies were carried out in the European north-east of Russia (Komi Republic) and Central Siberia (Krasnoyarsk region). The objects of the study were typical Glossic Stagnic Retisol (Siltic, Cutanic). The time after the fire was determined by dendrochronological methods. Data on the content of water-soluble organic matter and densimetric fractions of soils were obtained; carbon and nitrogen stocks were calculated. The content of polycyclic aromatic hydrocarbons (PAHs) was established to characterize the effect of fires. Pyrogenic carbonaceous inclusions were morphologically diagnosed 200 years after the fire. In this regard, it is proposed to distinguish a "pyrogenic" subtype for soils with pronounced signs of pyrogenesis. Carbon stocks in soils of the Komi Republic varied from 5.7 to 15.7 kg C m$^{-2}$, and soils of the Krasnoyarsk region had an accumulation of 6.9–12.5 kg C m$^{-2}$. The contribution of the pyrogenic horizon Epyr to the total carbon and nitrogen stocks was 9–45%. It is suggested that pyrogenic carbon (PyC) can accumulate in light densimetric fractions (fPOM$_{<1.6}$ and oPOM$_{<1.6}$). The analysis of PAH content showed their high concentrations in the organic and upper mineral horizons of the studied soils (24 to 605 ng g$^{-1}$). The coefficient FLA (fluoranthene)/(FLA+PYR(pyrene)) was the most useful to diagnose the pyrogenic origin of PAHs in the studied Retisols.

**Keywords:** boreal forest; PAHs; fire; soil organic matter; carbon and nitrogen stocks

## 1. Introduction

Boreal forests concentrate up to 60% of the planet's carbon in their biomes [1,2]. At the same time, the soils of boreal forests are the most important pool of soil organic matter, which contains up to 30% of the soil carbon of ecosystems [3]. It is known that there are no forests that are not prone to fires. It is estimated that approximately 1% of boreal forests are burned annually [4,5].

The boreal forests of the European North of Russia are characterized by the frequency of fires from 1–2 per century to 1–2 per millennium [6], with a maximum during the Holocene optimum [7]; in the forests of Siberia, the average interval between fires is from 50 to 320 years [8,9]. It is generally estimated [10] that fires in boreal forests produce 7–17 Tg of pyrogenic carbon per year. Its contribution can be from 1.6 to 60% of the total content of soil organic carbon [11,12].

Fires in boreal forests lead to a significant transformation of forest ecosystems [13,14], and pyrogenic carbon (PyC) is regarded as one of the most stable and sustainable pools of soil carbon [10,15]. Many experts note that forest fires change the chemical composition of soil organic matter (SOM) [16–19]. As a result of the fire, the reserves of organic matter change, the contribution of components with a high degree of decomposition [20] and

pyrogenically modified organic matter increases [15,21], and the content of compounds of aromatic nature also increases. During the pyrolysis of coniferous wood containing lignin, polycyclic aromatic hydrocarbons (PAHs) are formed. It is noted [22] that light PAHs are substances indicative of past fires, while the content of heavy PAHs indicates the intensity of the fire.

Despite a large number of studies assessing the impact of the pyrogenic factor on boreal forest ecosystems, there are practically no works devoted to the impact of fires on soils and soil organic matter in spruce communities. This is probably due to the fact that spruce forests burn to a lesser extent compared to pine and larch forests. A large number of research studying the changes in ecosystems and soils under the influence of fires have been conducted on the territory of Siberia [20,23–27]. In most cases, pyrogenic effects are studied in soils only in the first years after a fire [20,28]. Despite the widespread occurrence of pyrogenesis, the impact of a fire is practically not taken into account when analyzing the further functioning and development of forest areas. There are almost no works describing the long-term dynamics of post-pyrogenic communities and pyrogenic carbon. Analyzing post-fire succession is important to understand boreal forest dynamics and how it affects SOM. The forests of protected natural territories play a special role, serving as standards of the current state of boreal ecosystems, and are characterized by a high complexity of vegetation and soil cover.

In this study, we analyze the morphological and chemical properties of soils of old-age indigenous spruce forests of the Komi Republic and the Krasnoyarsk Territory affected by fire 100–200 years ago. It is assumed that the soils of the indigenous spruce forests of two large "forest" regions of Russia (the Krasnoyarsk Territory and the Komi Republic) have similar patterns of influence of pyrogenesis on the basic properties of soils. Due to this, they can be used as standards for studying the long-term impact of fires on boreal ecosystems.

## 2. Materials and Methods

### 2.1. Area Description

This work presents the results of a study of six soils formed in old spruce forests (*Picea obovata* Ledeb.). The spruce forests burned between 100 and 204 years ago. The study sites were located in geographically remote territories of the Komi Republic and the Krasnoyarsk region (Figure 1).

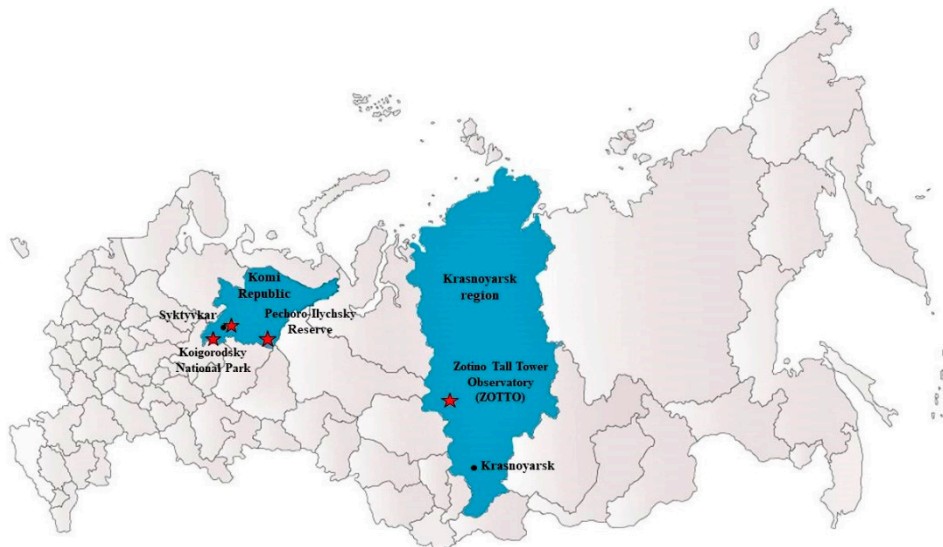

**Figure 1.** Location of the research area. The study sites are marked with stars.

The objects of the study in the Krasnoyarsk region were located in the vicinity of the ZOTTO International Observatory. Sites I-CS and II-CS were located on the left bank of the

Vorogovka River. Site III-CS was located on the right bank of the Porozhnaya River. In the Komi Republic, the I-NE site was located in the vicinity of the Koygorodsky National Park. The site II-NE was located on the territory of the Pechora-Ilych Reserve on the left bank of the Pechora River. The third site III-NE was located near Syktyvkar on the territory of the Maksimovsky monitoring station of the Institute of Biology of the Komi National Research Center of the Ural Branch of the Russian Academy of Sciences. For more information on the research objects, see Table 1 and Figure A1.

**Table 1.** Description of the objects of research.

| Site | Coordinates Height a.s.l., m | Time after Fire, Years | Vegetation | Soil Diagnostic Horizons |
|---|---|---|---|---|
| | | | Krasnoyarsk region | |
| I-CS | 60°56′23.3″ N 89°56′00.6″ E h = 165 | 146 | **Blueberry-green-moss spruce forest** **Grass-shrub layer** (TCP 40%–60%) *Vaccinium myrtillus, Gymnocarpium dryopteris, Linnaea borealis, Oxalis acetosella, Maianthemum bifolium, Lycopodium annotinum, Carex globularis, Calamagrostis obtusata.* **Moss-lichen layer** (TCP 90%) *Pleurozium schereberi u Hylocomium splendens, Polytrichum commune, Ptilium crista-castrensis, Sphagnum* sp. | Oi–O(e/a)$_{pyr}$–E$_{pyr}$–E– BE–Bt–BC |
| II-CS | 60°57′11.5″ N 89°53′06.6″ E h = 179 | 196 | **Fern-green moss spruce forest** **Grass-shrub layer** (TCP 50%–70%) *Gymnocarpium dryopteris, Vaccinium myrtillus, Calamagrostis purpurea, Linnaea borealis, Equisetum sylvaticum, Oxalis acetosella, Maianthemum bifolium, Lycopodium annotinum, Trientalis europaea, Lycopodium clavatum, Calamagrostis obtusata.* **Moss-lichen layer** (TCP 80%–90%) *Pleurozium schereberi, Polytrichum commune, Sphagnum girgensohnii, Hylocomium splendens.* | Oi–O(e/a)$_{pyr}$–E$_{pyr}$–E1– E2–BE–Bt–BC |
| III-CS | 61°04′07.7″ N 89°48′48.0″ E h = 142 | >100. | **Blueberry-green-moss spruce forest** **Grass-shrub layer** (TCP 50%) *Vaccinium myrtillus, Gymnocarpium dryopteris, Oxalis acetosella, Maianthemum bifolium, Linnaea borealis, Equisetum pratense, Pyrola rotundifolia, Calamagrostis obtusata, Lycopodium annotinum.* **Moss-lichen layer** (TCP 90%) *Pleurozium schreberi, Ptilium crista-castrensis, Hylocomium splendens, Polytrichum commune* | Oi–O(e/a)$_{pyr}$–E$_{pyr}$–E1– E2–BE–Bt |
| | | | Komi Republic | |
| I-EN | 59°58′52.5″ N 50°08′49.5″ E h = 168 | 140 | **Blueberry-sphagnum spruce forest** **Grass-shrub layer** (TCP 30%) *Vaccinium myrtillus, Vaccinium vitis-idaea, Trientalis europaea, Carex globularis, Maianthemum bifolium.* **Moss-lichen layer** (TCP 80%) *Sphagnum angustifolium, Sphagnum girgensohnii, Polytrichum commune, Pleurozium schreberi, Ptilium crista-castrensis.* | Oi–Oe–Oa–E$_{pyr}$–E2– BE–Bt |

**Table 1.** *Cont.*

| Site | Coordinates Height a.s.l., m | Time after Fire, Years | Vegetation | Soil Diagnostic Horizons |
|---|---|---|---|---|
| | | | Komi Republic | |
| II-EN | 62°03′49.5″ N 58°27′02.6″ E h = 210 | 204 | **Blueberry-green-moss spruce forest** **Grass-shrub layer** (TCP 60%) *Vaccinium myrtillus, Equisetum sylvaticum, Carex globularis, Lycopodium annotinum, Linnaea borealis, Dryopteris expansa.* **Moss-lichen layer** (TCP 95%) *Polytrichum commune, Sphagnum girgensohnii, Sphagnum balticum, Pleurozium schreberi, Hylocomium splendens* | Oi–Oe–Oa–$E_{pyr}$–E–BE– BE2–Bt–BC |
| III-EN | 61°39′45.2″ N 50°41′09.1″ E h = 151 | 100 | **Blueberry-green-moss spruce forest** **Grass-shrub layer** (TCP 70%) *Vaccinium myrtillus, Rubis saxatilis, Stellaria longifolia, Mainathemum bifolium, Oxalis acetosella.* **Moss-lichen layer** (TCP 95%) *Pleurozium schreberi, Climacium dendroides, Polytrichum commune* | Oi–Oe–Oa–$E_{pyr}$–E1– E2–BE–Bt–BC |

Note: EN—European North (Komi Republic); CS—Central Siberia (Krasnoyarsk region); I, II, III—site numbers; TPC—total projective cover.

*2.2. Dendrochronology*

Wood samples were taken from living and dead wood [29]. All the collected material was cleaned for its dating. For clearer visibility of tree rings and fire scars, all samples were cleaned with a blade and then covered with tooth powder. The dating of fires on the prepared wood samples was carried out using the CDendro 9.0.1 and CooRecorder 9.0.1 programs (Cybis Elektronik & Data AB, Saltsjöbaden, Sweden). The tree ring width was measured with an accuracy of 0.01 mm on a LINTAB semiautomatic device under a binocular microscope with 40× magnification using the TsapWin computer program [30]. A number of chronologies were built on living trees to identify old fires. After that, the date of the fire calculated from the dead trees was determined using the cross-dating method. To determine the accuracy of the data, the data collected were checked using cross-correlation analysis using the COFECHA software [31,32].

*2.3. General Soil Analyses*

Samples were selected for chemical analyses from each genetic horizon. The color of the horizon was determined by Munsell Soil Color Charts [33]. Samples were air-dried and living visible roots and all particles with a diameter >2 mm were removed by dry sieving. Organic layer samples were ground with a ball mill.

The chemical analyses of the soils were performed in the certified Ecoanalit laboratory of the Institute of Biology (Komi Science Center, Urals Branch of the Russian Academy of Sciences) (certificate ROSS RU.0001.511257 from September 2019). The pH values were determined on an HI2002-02 edge series pH meter with a Hanna digital pH electrode (Hanna Instruments, Romania) at a soil:water ratio of 1 (m):2.5 (v) for mineral horizons and 1 (m):25 (v) for organic and pyrogenic horizons. Exchangeable cations ($Ca^{2+}$, $Mg^{2+}$, $K^+$, and $Na^+$) were extracted by 1M $CH_3COONH_4$ on a mechanical extractor (Sampletekk, Mav co Industries Inc., Lawrenceburg, IN, USA). The granulometric composition of soils was determined according to [34]. The classification of the soil was determined using the Ferre triangle. The measurement of dithionite-soluble iron was performed with a modified method according to Mehra and Jackson [35]. The bulk density (BD) of organic and mineral horizons was determined according to [36].

### 2.4. Soil Organic Matter Analysis

Total C and N were determined by dry combustion on an EA-1100 analyzer (Carlo Erba, Milano, Italy). Due to low pH and the negligible contents of mineral N in the forest floor, total C and N were considered to represent organic C and N. Carbon and nitrogen pools were calculated using the formula according to Hiederer and Köchy [37]:

$$C_{stock} \left( kg\ m^{-2} \right) = 0.1 \times C_{tot} \times BD \times h$$

where h is the thickness of the soil layer (cm), $C_{tot}$ is the organic carbon concentration of the soil layer (g kg$^{-1}$), and *BD* is the bulk density of the soil layer (kg m$^{-3}$).

Water-soluble organic carbon (WSOC) and nitrogen (WSON) were extracted with deionized water (ELGA Lab Water, England) at room temperature at a ratio of 1:50 (soil: water) for mineral horizons and 1:100 for organic and pyrogenic horizons in BIOFIL test tubes. Suspensions were shaken for an hour using the Heidolph Multi Reax shaker (acceleration 6×) at room temperature. Filtration was performed immediately after shaking and Millipore devices with quartz filters were utilized (MN, Düren, Germany; pore size 0.4 μm). Each sample was measured for the total volume of filtrate. Total carbon (TC) and nitrogen (TN) were assessed using the TOC-VCPN analyzer (Shimadzu, Japan) with TNM-1 module. The obtained results were recalculated on a dry soil mass basis.

The upper pyrogenic horizons of $E_{pyr}$ were studied by the method of densimetric fractionation. Densimetric fractionation was performed according to the method proposed by [38] with modifications [39]. We used a sodium polytungstate (SPT0) solution with a density of $1.60 \pm 0.03$ g cm$^{-3}$ for density fractionations. Three fractions of organic matter were isolated: free particulate organic matter (fPOM$_{<1.6}$), occluded particulate organic matter (oPOM$_{<1.6}$), and heavy fraction associated with the mineral matrix (MaOM$_{>1.6}$).

To determine PAHs, the $O_{pyr}$ and $E_{pyr}$ horizons were studied. The extraction of PAHs from the soils was made on an Accelerated Solvent Extractor 350 (Thermo Fischer Scientific™, Waltham, MA, USA) in the Chromatography Collective Use Center of the IB FRC Komi SC UrB RAS. The content of PAHs in the concentrates was determined by reversed-phase high-performance liquid chromatography in a gradient mode with a spectrofluorimetric detection. The extraction was fulfilled three times by a dichlormethane:acetone (1:1) mixture at 100 °C. The extracts of 2 cm$^3$ in volume were purified from organic impurities according to the US EPA purification method (1986) by means of the column chromatography using activated aluminum oxide (Brockmann II grade) (Fluka, cat.no. 06300; particle size 0.05–0.15 mm, activated at 600 °C for 4 h, and partially deactivated with 3% H$_2$O). A 0.5 cm layer of sodium sulfate was added to the top of the column. The hexane:methylene chloride (4:1) mixture (30 cm$^3$) was applied as an eluent. The eluates were concentrated using a Kuderna Danish concentrator at the thermostat temperature of 90 °C to a volume of 1–2 cm$^3$. Then, 3 cm$^3$ of acetonitrile were added to them and these mixtures were evaporated at 90 °C until a complete removal of hexane. The US EPA method 8310 (1986) and certified national standard method of quantitative chemical analysis (PND F 16.1:2.2:2.3:3.62-09, 2009) were selected for PAH measurement.

### 2.5. Statistics

Correlation coefficients (*r*-Pearson) were calculated using the STATISTICA 10.0 (Stat. Soft Inc., Tusla, OK, USA); differences were considered significant at the significance level of $p < 0.05$.

## 3. Results

### 3.1. Dendrochronology

The close time intervals after the fire in the spruce communities of the Krasnoyarsk Territory and the Komi Republic were revealed in terms of the frequency of fires. In Siberian spruce forests, this interval is 100–196 years. In the Komi Republic, the post-fire interval varies from 100 to 204 years. The regularity of the frequency of post-fire dynamics is clearly

traced, depending on the distance from settlements. Thus, on the site III-EN located near Syktyvkar, the post-fire period was 100 years. At the I-EN point, a new forest plantation was fully formed after the fire. This point is also located near a large number of settlements and most likely the fire occurred as a result of the human factor. The most inaccessible point, II-EN, is located in the foothills of the Northern Ural. The fire at this point had a local character, as a survey of nearby spruce forests showed the visual absence of coal in the soil. The spruce forests of Siberia are located at a considerable distance from populated areas. However, due to their high continentality, the frequency of fires and the dissemination area of fire were much higher, unlike the spruce forests of the European part.

### 3.2. Morphological Soil Properties

Photos of study sites are shown in Appendix A. The organic horizon of soils consists of several sub-horizons, which are plant residues at different degrees of decomposition: Oi, Oe, and Oa. In some cases, the O(e/a) sub-horizon is isolated. The upper sub-horizon Oi with a thickness from 0 to 3 cm is formed from mosses and the previous year's needles and leaves, and is a slightly decomposed plant material. The Oe sub-horizon was isolated only for the soils of the Komi Republic, and this sub-horizon was not detected for the soils of the Krasnoyarsk region. The organic horizon Oe is characterized as a medium-decomposed organic material with a thickness of 3–8 cm from the remains of mosses and shrubs densely intertwined with roots. The lower Oa or O(e/a) sub-horizon with a thickness of 2–5 cm is a well-decomposed plant organic material, the darkest in the soil profile. Coal particles of various sizes are found in the horizon, which is an important diagnostic sign for assessing the impact of a fire.

The mineral profile of soils has a precise differentiation into genetic horizons. In the upper part of the profile, a horizon E light loam with a thickness of up to 55 cm, whitish in color, is formed (2.5 5/2-5/3-6/2-6/3, 10YR5-3-5/4 on Munsell). The horizon can also be divided into several sub-horizons that differ in morphological features. In the upper part of the mineral profile, the $E_{pyr}$ horizons of small thickness (4–13 cm) are formed, which are characterized by the presence of coal particles and impregnation with organic matter. The presence of coal particles in the mineral horizons indicates the passage of the forest stand by fire. The $E_{pyr}$ horizon of darker color (10YR 3/3-4/3-5/3) and structureless as opposed to the underlying E horizons. Under the E horizon, a transitional weakly BE horizon of light brown color (2.5 4/4-5/3, 10YR 4/6-5/4) with a thickness between 10 and 25 cm is formed. Below, the horizon of the Bt is formed. The horizon has a brown color (7.5YR4/3, 10YR 4/3-5/4) and is a heavy loam, which is characterized by a well-defined structure.

### 3.3. Basic Chemical Properties of Soils

The main results of chemical analyses are presented in Table 2. The soils of the studied native spruce forests of the Komi Republic and the Krasnoyarsk region are characterized by similar chemical properties. The $pH_{H2O}$ of soils varies from strongly acidic to neutral values in organic horizons (4.1–6.0). Mineral horizons are less acidic, as $pH_{H2O}$ values gradually increase down the profile (up to 7.3). Among the mineral horizons, the upper horizons E (4.1–4.7) are the most acidic.

The analysis of the exchangeable cation content showed the predominance of $Ca^{2+}$ in the studied soils. The content of $Mg^{2+}$, $K^+$, and $Na^+$ cations was lower. Organic horizons are characterized by maximum values of exchangeable cations. The distribution of the cation content is characterized by a sharp decrease in the mineral horizons E and BE. An increase in the content of Bt and BC in the lower mineral horizons was revealed. The base saturation (Bs) and the cation exchange capacity (CEC) have a similar type of distribution with the content of exchangeable cations. Organic horizons (up to 96%) and lower mineral horizons of the studied soils (up to 131%) are the most saturated with cations.

**Table 2.** Chemical properties of the studied soils.

| Site | Soil Horizon | Depth, cm | pH H$_2$O | pH KCl | Ca$^{2+}$ | Mg$^{2+}$ | K$^+$ | Na$^+$ | $\Sigma$ | CEC | BS | Al$_{ox}$ | Fe$_{ox}$ | Fe$_{dith}$ | Ks |
|------|------|------|------|------|------|------|------|------|------|------|------|------|------|------|------|
| | | | | | cmol kg$^{-1}$ | | | | | | | % | | | |
| | | | | | | Krasnoyarsk region | | | | | | | | | |
| I-CS | Oi | 0–2 | 5.4 | 4.6 | 23.8 | 8.8 | 10.7 | 0.07 | 43.4 | 66.2 | 66 | – | – | – | – |
| | O(e/a)$_{pyr}$ | 2–5 | 4.3 | 3.1 | 12.4 | 3.5 | 2.1 | 0.05 | 18.0 | 79.4 | 23 | – | – | – | – |
| | E$_{pyr}$ | 5–18 | 4.1 | 3.2 | 0.4 | 0.4 | 0.3 | 0.03 | 1.0 | 21.2 | 5 | 0.54 ± 0.13 | 0.62 ± 0.09 | 0.96 | 0.6 |
| | E | 18–30 | 5.1 | 3.8 | 0.9 | 0.7 | 0.2 | 0.02 | 1.8 | 11.3 | 16 | 0.63 ± 0.15 | 0.63 ± 0.09 | 1.09 | 0.6 |
| | BE | 30–55 | 5.2 | 3.8 | 1.1 | 0.9 | 0.2 | 0.02 | 2.2 | 9.5 | 24 | 0.51 ± 0.12 | 0.50 ± 0.08 | 1.02 | 0.5 |
| | Bt | 55–75 | 5.3 | 3.6 | 2.8 | 1.8 | 0.2 | 0.03 | 4.9 | 12.2 | 40 | 0.45 ± 0.11 | 0.34 ± 0.05 | 1.16 | 0.3 |
| | BC | 75–90 | 5.6 | 3.5 | 6.7 | 3.0 | 0.2 | 0.03 | 9.9 | 13.0 | 77 | 0.27 ± 0.07 | 0.52 ± 0.08 | 1.67 | 0.3 |
| II-CS | Oi | 0–2 | 5.1 | 4.3 | 22.8 | 8.6 | 9.9 | 0.06 | 41.4 | 70.5 | 59 | – | – | – | – |
| | O(e/a)$_{pyr}$ | 2–5 | 4.4 | 3.3 | 8.4 | 3.6 | 2.0 | 0.13 | 14.2 | 88.5 | 16 | – | – | – | – |
| | E$_{pyr}$ | 5–10 | 4.3 | 3.3 | 0.5 | 0.3 | 0.1 | 0.02 | 0.9 | 15.2 | 6 | 0.50 ± 0.12 | 0.64 ± 0.10 | 0.79 | 0.8 |
| | E | 10–35 | 5.1 | 3.6 | 1.0 | 0.7 | 0.1 | 0.03 | 1.9 | 10.1 | 19 | 0.37 ± 0.09 | 0.68 ± 0.10 | 0.86 | 0.8 |
| | E2 | 35–60 | 5.5 | 3.6 | 2.7 | 1.6 | 0.1 | 0.05 | 4.5 | 10.3 | 43 | 0.31 ± 0.07 | 0.67 ± 0.10 | 0.76 | 0.9 |
| | BE | 60–75 | 5.9 | 3.7 | 8.6 | 4.9 | 0.3 | 0.08 | 13.9 | 17.4 | 79 | 0.36 ± 0.09 | 0.55 ± 0.08 | 0.78 | 0.7 |
| | Bt | 75–90 | 6.2 | 3.9 | 14.2 | 7.7 | 0.4 | 0.13 | 22.4 | 22.6 | 99 | 0.36 ± 0.09 | 0.52 ± 0.08 | 0.91 | 0.6 |
| | BC | 90–110 | 6.3 | 4.0 | 15.6 | 8.5 | 0.4 | 0.14 | 24.7 | 24.1 | 102 | 0.37 ± 0.09 | 0.58 ± 0.09 | 0.97 | 0.6 |
| III-CS | Oi | 0–2 | 5.0 | 4.3 | 22.9 | 5.8 | 10.0 | 0.04 | 38.8 | 65.0 | 60 | – | – | – | – |
| | O(e/a)$_{pyr}$ | 2–5 | 4.5 | 3.3 | 14.3 | 3.3 | 3.3 | 0.04 | 20.9 | 79.4 | 26 | – | – | – | – |
| | E$_{pyr}$ | 5–17 | 4.6 | 3.5 | 1.0 | 0.5 | 0.2 | 0.01 | 1.7 | 16.2 | 11 | 0.55 ± 0.13 | 0.86 ± 0.13 | 1.07 | 0.8 |
| | E1 | 17–30 | 5.2 | 3.7 | 1.5 | 0.9 | 0.1 | 0.02 | 2.5 | 12.3 | 20 | 0.46 ± 0.11 | 0.71 ± 0.11 | 0.96 | 0.7 |
| | E2 | 30–50 | 5.3 | 3.6 | 2.4 | 1.5 | 0.1 | 0.03 | 4.1 | 11.1 | 37 | 0.35 ± 0.08 | 0.56 ± 0.08 | 0.86 | 0.6 |
| | BE | 50–70 | 6.0 | 3.7 | 11.1 | 7.2 | 0.4 | 0.06 | 18.7 | 21.7 | 86 | 0.42 ± 0.10 | 0.43 ± 0.06 | 1.04 | 0.4 |
| | Bt | 70–90 | 7.3 | 6.1 | 19.0 | 12.0 | 0.4 | 0.06 | 31.6 | 24.2 | 131 | 0.35 ± 0.08 | 0.43 ± 0.06 | 1.24 | 0.3 |
| | | | | | | Komi Republic | | | | | | | | | |
| I-EN | Oi | 0–3 | 4.6 | 3.4 | 15.7 | 6.5 | 14.3 | 0.33 | 36.9 | 46.7 | 79 | – | – | – | – |
| | Oe | 3–11 | 4.1 | 3.0 | 7.1 | 2.7 | 2.1 | 0.22 | 12.1 | 96.7 | 12 | – | – | – | – |
| | Oa$_{pyr}$ | 11–16 | 4.4 | 3.5 | 2.6 | 1.3 | 1.7 | 0.14 | 5.8 | 130.0 | 4 | – | – | – | – |
| | E$_{pyr}$ | 16–30 | 4.8 | 3.8 | 0.2 | 0.1 | 0.1 | 0.01 | 0.3 | 8.6 | 4 | 0.25 ± 0.06 | 0.34 ± 0.11 | 0.32 | 1.1 |
| | E2 | 30–50 | 5.2 | 4.0 | 0.3 | 0.2 | 0.1 | 0.02 | 0.6 | 7.2 | 8 | 0.30 ± 0.07 | 0.64 ± 0.10 | 0.74 | 0.9 |
| | BE | 50–75 | 5.4 | 3.9 | 2.3 | 1.1 | 0.2 | 0.03 | 3.5 | 11.0 | 32 | 0.34 ± 0.08 | 0.54 ± 0.08 | 0.84 | 0.6 |
| | Bt | 75–90 | 5.7 | 3.8 | 10.3 | 4.6 | 0.4 | 0.07 | 15.3 | 20.9 | 73 | 0.38 ± 0.09 | 0.43 ± 0.06 | 0.98 | 0.4 |
| II-EN | Oi | 0–2 | 5.1 | 4.2 | 19.4 | 6.6 | 11.6 | 0.13 | 37.8 | 70.0 | 54 | – | – | – | – |
| | Oe | 2–6 | 4.2 | 3.1 | 11.7 | 2.7 | 2.2 | 0.12 | 16.8 | 78.0 | 22 | – | – | – | – |
| | Oa$_{pyr}$ | 6–8 | 4.1 | 3.0 | 5.1 | 1.7 | 1.3 | 0.15 | 8.2 | 90.9 | 9 | – | – | – | – |
| | E$_{pyr}$ | 8–12 | 4.2 | 3.4 | 0.3 | 0.3 | 0.3 | 0.09 | 1.0 | 21.7 | 5 | 0.54 ± 0.13 | 0.40 ± 0.14 | 0.51 | 0.8 |
| | E | 12–20 | 4.4 | 3.6 | 0.2 | 0.2 | 0.2 | 0.06 | 0.6 | 23.7 | 3 | 0.69 ± 0.16 | 0.85 ± 0.13 | 0.97 | 0.9 |
| | BE | 20–35 | 4.9 | 4.0 | 0.4 | 0.4 | 0.4 | 0.04 | 1.2 | 14.0 | 9 | 0.65 ± 0.15 | 0.60 ± 0.09 | 0.92 | 0.7 |
| | BE2 | 35–55 | 5.2 | 4.0 | 2.9 | 1.9 | 0.7 | 0.21 | 5.8 | 37.2 | 16 | 0.63 ± 0.15 | 0.57 ± 0.09 | 1.14 | 0.5 |
| | Bt | 55–80 | 5.6 | 3.9 | 13.0 | 6.9 | 0.8 | 0.25 | 20.9 | 39.6 | 53 | 0.51 ± 0.12 | 0.45 ± 0.07 | 0.95 | 0.5 |
| | BC | 80–100 | 6.1 | 4.1 | 9.0 | 4.3 | 0.3 | 0.10 | 13.7 | 15.2 | 90 | 0.29 ± 0.07 | 0.30 ± 0.10 | 1.00 | 0.3 |
| III-EN | Oi | 0–1 | 6.0 | 5.4 | 51.9 | 14.8 | 7.3 | 0.16 | 74.2 | 77.3 | 96 | – | – | – | – |
| | Oe | 1–4 | 5.3 | 4.6 | 38.7 | 5.1 | 3.7 | 0.12 | 47.7 | 78.5 | 61 | – | – | – | – |
| | Oa | 4–6 | 4.8 | 3.9 | 18.8 | 2.0 | 1.3 | 0.18 | 22.3 | 76.8 | 29 | – | – | – | – |
| | E$_{pyr}$ | 6–10 | 4.7 | 3.7 | 1.6 | 0.2 | 0.1 | 0.06 | 2.0 | 7.9 | 25 | 0.17 ± 0.04 | 0.14 ± 0.05 | 0.19 | 0.7 |
| | E1 | 10–25 | 5.2 | 4.4 | 0.4 | 0.1 | 0.1 | 0.02 | 0.6 | 5.8 | 10 | 0.36 ± 0.09 | 0.63 ± 0.09 | 0.74 | 0.8 |
| | E2 | 25–35 | 5.6 | 4.1 | 1.4 | 0.5 | 0.1 | 0.05 | 2.0 | 3.5 | 57 | 0.14 ± 0.03 | 0.33 ± 0.05 | 0.47 | 0.7 |
| | BE | 35–45 | 5.7 | 3.8 | 5.2 | 2.1 | 0.2 | 0.07 | 7.6 | 11.4 | 66 | 0.25 ± 0.06 | 0.41 ± 0.06 | 0.78 | 0.5 |
| | Bt | 45–70 | 5.9 | 4.0 | 12.1 | 5.0 | 0.5 | 0.14 | 17.7 | 17.5 | 101 | 0.31 ± 0.07 | 0.31 ± 0.05 | 0.97 | 0.3 |
| | BC | 70–100 | 6.2 | 4.2 | 11.9 | 4.9 | 0.4 | 0.10 | 17.4 | 17.3 | 100 | 0.24 ± 0.06 | 0.024 ± 0.08 | 0.93 | 0.3 |

Note: EN—European North (Komi Republic); CS—Central Siberia (Krasnoyarsk region); I, II, III—site numbers; CEC—cation exchange capacity; BS—base saturation; Ks—Schwertmann coefficient.

The analysis of the granulometric composition (Table A1) showed the predominance of silt and clay fractions in the studied soils. It is established that the soils of the Komi

Republic have a heavier mechanical composition than the soils of the Krasnoyarsk region. Nevertheless, the studied soils are characterized by similar patterns of distribution of mechanical fractions in soil horizons.

The content of sand fractions in the soils of Komi Republic (2.000–0.050 mm) varied from 0 to 17%, and silt fractions (0.050–0.002 mm) from 6 to 33%. The clay fraction (<0.002 mm) prevailed in the granulometric composition. Its content was 32%–65%. In the soils of Krasnoyarsk region, the contents of sand, silt, and clay fractions were 0%–13%, 2%–16%, and 19%–51%, respectively. The determination of the mechanical composition of the soil by the Ferre method revealed that the mineral horizons of the studied soils belong to various types of loam.

The content of $Fe_{dith}$ in the soils of the Komi Republic varied from 0.19 to 1.14% in mineral horizons. In the soils of the Krasnoyarsk region, higher values of $Fe_{dith}$ were found —from 0.79 to 1.67%. In the soils of the Krasnoyarsk region, $Fe_{ox}$ was in the range of 0.27%–0.86%; in the soils of the Komi Republic, similar values were found, from 0.24%–0.85%. The content of $Al_{ox}$ was also similar in the soils of the Komi Republic (0.14%–0.69%) and Krasnoyarsk region (0.27%–0.63%).

The degree of soil hydromorphism can be determined according to the Schwertmann coefficient: $Ks = F_{ox}/Fe_{dith}$ [40]. It is established that the soils of the studied regions are quite close to each other by the nature of hydromorphism. The Schwertmann coefficient for the soils of the Komi Republic is $Ks = 0.3–1.1$, and for the soils of the Krasnoyarsk Territory, $Ks = 0.3–0.9$. The upper mineral horizons were characterized by a high water content. The obtained $Ks$ values are confirmed by the actual moisture contents of the genetic horizons of the studied soils (Figure 2A). The water content of organic horizons of the soils of the Komi Republic varies from 264.8 to 397.5%, and in the Krasnoyarsk region 167.2%–248.6%. Among the mineral horizons, the wettest are horizons E (23.2%–44.0%). Water content indicators in the lower BE and Bt horizons were 17.5%–28.4% and 20.2%–22.8%, respectively.

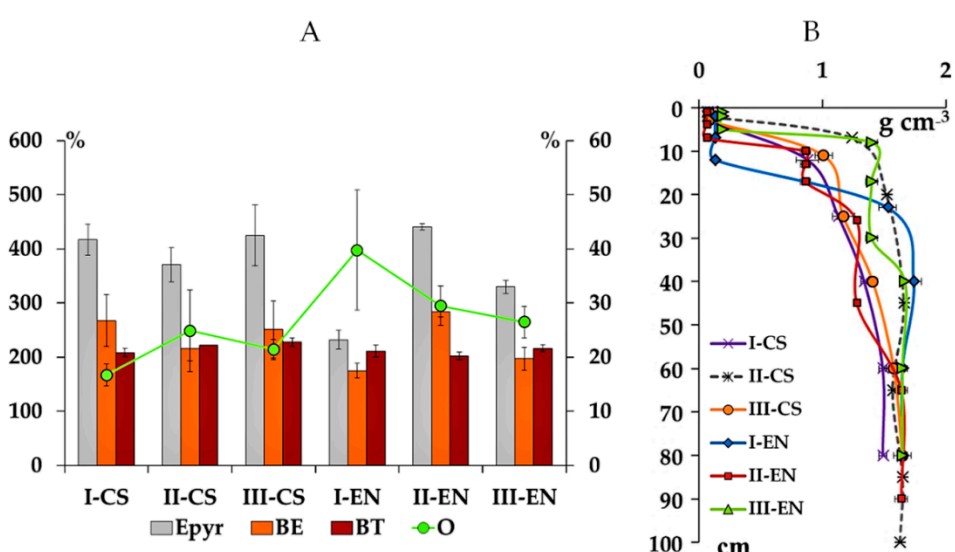

**Figure 2.** Water contents of organic ((**A**) left axis) and mineral ((**A**) right axis) horizons and bulk density (**B**) of the studied soils. EN—European North (Komi Republic); CS—Central Siberia (Krasnoyarsk region); I, II, III—site numbers.

An important morphological indicator of soils is bulk density (Figure 2B). The minimum density of addition is typical for organic horizons; the maximum values are for mineral horizons. The organic horizons of the Komi Republic soils are characterized by a density equal to 0.06–0.19 g cm$^{-3}$, and mineral horizons 0.86–1.74 g cm$^{-3}$. In the soils of the Krasnoyarsk region, the density of organic horizons was lower and amounted to 0.07–0.09 g cm$^{-3}$, and the density of mineral horizons varied from 0.88 to 1.66 g cm$^{-3}$.

### 3.4. Content of Carbon and Nitrogen

The maximum concentrations of carbon ($C_{tot}$) and nitrogen ($N_{tot}$) were found for the organic horizons of the studied soils (Table 3). The carbon content in the organic horizons of the soils of the Komi Republic varied from 22.0 to 47.0%, and nitrogen 0.68%–1.73%. In the organic horizons of the soils of the Krasnoyarsk region, the carbon content varied in the range of 37.3%–48.0%, and the nitrogen content 1.13%–1.71%. The content of elements in mineral horizons is significantly reduced. The carbon content in the mineral horizons of the soils of the Komi Republic was 0.14 to 3.0%; the nitrogen content was 0.014%–0.15%. The mineral horizons of the soils of the Krasnoyarsk region contained 0.26%–2.7% of total carbon and 0.030%–0.123% of total nitrogen.

**Table 3.** Carbon and nitrogen contents in the studied soils.

| Site | Soil Horizon | Depth. cm | $C_{tot}$ | $N_{tot}$ | C/N | WSOC | WSON | C/N$_{WS}$ | $C_{stock}$ | $N_{stock}$ |
|---|---|---|---|---|---|---|---|---|---|---|
| | | | g kg$^{-1}$ | | | mg g$^{-1}$ | | | kg m$^{-2}$ | |
| | | | | | Krasnoyarsk region | | | | | |
| I-CS | Oi | 0–2 | 373 ± 13 | 11.3 ± 1.2 | 39 | 10.31 | 0.82 | 15 | 0.7 | 0.02 |
| | O(e/a)$_{pyr}$ | 2–5 | 424 ± 15 | 14.8 ± 1.6 | 33 | 3.55 | 0.18 | 23 | 1.1 | 0.04 |
| | E$_{pyr}$ | 5–18 | 48 ± 7 | 2.3 ± 0.5 | 24 | 0.67 | 0.02 | 32 | 5.5 | 0.26 |
| | E | 18–30 | 8.5 ± 2 | 0.58 ± 0.12 | 17 | 0.21 | 0.01 | 36 | 1.2 | 0.08 |
| | BE | 30–55 | 5.2 ± 1.2 | 0.44 ± 0.09 | 14 | 0.18 | 0.005 | 44 | 1.7 | 0.15 |
| | Bt | 55–75 | 3.1 ± 0.7 | 0.39 ± 0.11 | 9 | 0.16 | 0.003 | 52 | 0.9 | 0.12 |
| | BC | 75–90 | 2.7 ± 0.6 | 0.44 ± 0.09 | 7 | 0.14 | 0.003 | 58 | 1.0 | 0.16 |
| | | | | | | | | | 12.1 */15 **/45 *** | 0.83/7/31 |
| II-CS | Oi | 0–2 | 480 ± 17 | 14.2 ± 1.6 | 39 | 15.73 | 0.87 | 21 | 0.7 | 0.02 |
| | O(e/a)$_{pyr}$ | 2–5 | 398 ± 14 | 17.1 ± 1.9 | 27 | 5.19 | 0.24 | 25 | 0.8 | 0.04 |
| | E$_{pyr}$ | 5–10 | 16±4 | 0.81±0.16 | 23 | 0.32 | 0.01 | 44 | 1.0 | 0.05 |
| | E1 | 10–35 | 3.8±0.9 | 0.35±0.10 | 13 | 0.17 | 0.003 | 64 | 1.4 | 0.13 |
| | E2 | 35–60 | 2.8±0.6 | 0.30±0.09 | 11 | 0.16 | 0.003 | 62 | 1.2 | 0.12 |
| | BE | 60–75 | 2.6±0.6 | 0.34±0.10 | 9 | 0.15 | 0.002 | 72 | 0.6 | 0.08 |
| | Bt | 75–90 | 2.7±0.6 | 0.37±0.11 | 9 | 0.13 | 0.005 | 31 | 0.7 | 0.09 |
| | BC | 90–110 | 3.1 ± 0.7 | 0.44 ± 0.09 | 8 | 0.15 | 0.003 | 59 | 0.5 | 0.07 |
| | | | | | | | | | 6.9/22/15 | 0.61/9/8 |
| III-CS | Oi | 0–2 | 460 ± 16 | 12.3 ± 1.4 | 44 | 16.64 | 0.63 | 31 | 0.7 | 0.02 |
| | O(e/a)$_{pyr}$ | 2–5 | 399 ± 14 | 14.3 ± 1.6 | 33 | 6.24 | 0.34 | 21 | 1.0 | 0.03 |
| | E$_{pyr}$ | 5–17 | 27 ± 4 | 1.23 ± 0.25 | 26 | 0.47 | 0.01 | 42 | 3.3 | 0.15 |
| | E1 | 17–30 | 7.6 ± 1.8 | 0.48 ± 0.10 | 18 | 0.20 | 0.004 | 53 | 1.2 | 0.07 |
| | E2 | 30–50 | 4.5 ± 1.0 | 0.39 ± 0.11 | 13 | 0.18 | 0.004 | 57 | 1.3 | 0.11 |
| | BE | 50–70 | 3.7 ± 0.9 | 0.46 ± 0.09 | 9 | 0.14 | 0.003 | 63 | 1.2 | 0.15 |
| | Bt | 70–90 | 7.9 ± 1.8 | 0.50 ± 0.10 | 18 | 0.13 | 0.003 | 49 | 3.9 | 0.25 |
| | | | | | | | | | 12.5/14/26 | 0.78/7/19 |
| | | | | | Komi Republic | | | | | |
| I-EN | Oi | 0–3 | 366±13 | 11.1±1.2 | 38 | 12.00 | 0.70 | 20 | 1.4 | 0.04 |
| | Oe | 3–11 | 470±16 | 13.2±1.5 | 42 | 3.28 | 0.25 | 15 | 4.9 | 0.14 |
| | Oa$_{pyr}$ | 11–16 | 281±28 | 10.0±1.1 | 33 | 1.78 | 0.08 | 25 | 1.8 | 0.07 |
| | E$_{pyr}$ | 16–30 | 18±4 | 0.80±0.16 | 26 | 0.18 | 0.02 | 12 | 3.9 | 0.17 |
| | E2 | 30–50 | 4.7 ± 1.1 | 0.48 ± 0.10 | 11 | 0.12 | 0.01 | 13 | 1.6 | 0.17 |
| | BE | 50–75 | 2.6 ± 0.6 | 0.41 ± 0.08 | 7 | 0.10 | 0.01 | 9 | 1.1 | 0.17 |
| | Bt | 75–90 | 2.3 ± 0.5 | 0.40 ± 0.12 | 7 | 0.08 | 0.01 | 7 | 0.9 | 0.17 |
| | | | | | | | | | 15.7/52/25 | 0.92/27/19 |
| II-EN | Oi | 0–2 | 415 ± 15 | 14.3 ± 1.6 | 34 | 7.79 | 0.45 | 20 | 0.5 | 0.02 |
| | Oe | 2–6 | 447 ± 16 | 17.3 ± 1.9 | 30 | 2.75 | 0.17 | 19 | 1.1 | 0.04 |
| | Oa$_{pyr}$ | 6–8 | 421 ± 15 | 6.8 ± 0.7 | 72 | 1.74 | 0.09 | 22 | 0.5 | 0.01 |
| | E$_{pyr}$ | 8–12 | 30 ± 4 | 1.5 ± 0.3 | 23 | 0.36 | 0.01 | 44 | 1.0 | 0.05 |
| | E | 12–20 | 28 ± 4 | 1.46 ± 0.29 | 22 | 0.31 | 0.01 | 40 | 1.3 | 0.07 |
| | BE | 20–35 | 5.0 ± 1.1 | 0.49 ± 0.10 | 12 | 0.18 | 0.01 | 39 | 1.0 | 0.09 |
| | BE2 | 35–55 | 3.4 ± 0.8 | 0.42 ± 0.08 | 9 | 0.08 | 0.003 | 28 | 0.9 | 0.11 |
| | Bt | 55–80 | 2.0 ± 0.5 | 0.36 ± 0.10 | 6 | 0.06 | 0.003 | 18 | 0.8 | 0.15 |
| | BC | 80–100 | 1.4 ± 0.3 | 0.29 ± 0.08 | 6 | 0.05 | 0.004 | 14 | 0.5 | 0.10 |
| | | | | | | | | | 7.5/28/13 | 0.64/11/8 |

**Table 3.** *Cont.*

| Site | Soil Horizon | Depth. cm | $C_{tot}$ | $N_{tot}$ | C/N | WSOC | WSON | $C/N_{WS}$ | $C_{stock}$ | $N_{stock}$ |
|---|---|---|---|---|---|---|---|---|---|---|
| | | | g kg$^{-1}$ | | | mg g$^{-1}$ | | | kg m$^{-2}$ | |
| III-EN | Oi | 0–1 | $220 \pm 22$ | $10.0 \pm 1.1$ | 26 | 13.14 | 0.64 | 24 | 0.4 | 0.02 |
| | Oe | 1–4 | $271 \pm 27$ | $13.6 \pm 1.5$ | 23 | 8.92 | 0.48 | 22 | 1.5 | 0.08 |
| | Oa | 4–6 | $233 \pm 23$ | $7.3 \pm 0.8$ | 37 | 3.02 | 0.15 | 23 | 0.9 | 0.03 |
| | $E_{pyr}$ | 6–10 | $9.1 \pm 2.1$ | $0.67 \pm 0.13$ | 16 | 0.28 | 0.01 | 29 | 0.5 | 0.04 |
| | E1 | 10–25 | $3.9 \pm 0.9$ | $0.26 \pm 0.08$ | 18 | 0.11 | 0.003 | 38 | 0.8 | 0.05 |
| | E2 | 25–35 | <1.0 | <0.01 | - | 0.04 | – | – | 0.1 | 0.01 |
| | BE | 35–45 | <1.0 | $0.14 \pm 0.04$ | - | 0.04 | – | – | 0.2 | 0.02 |
| | Bt | 45–70 | $1.6 \pm 0.4$ | $0.26 \pm 0.08$ | 7 | 0.04 | – | – | 0.7 | 0.11 |
| | BC | 70–100 | $1.16 \pm 0.27$ | $0.23 \pm 0.07$ | 6 | 0.04 | – | – | 0.6 | 0.11 |
| | | | | | | | | | 5.7/50/9 | 0.47/26/8 |

Note: EN—European North (Komi Republic); CS—Central Siberia (Krasnoyarsk region); I, II, III—site numbers; WSOC—water-soluble organic carbon; WSON—water-soluble organic nitrogen; *—pools in the upper 100 cm; **—contribution of the organic horizon (%); ***—contribution of the $E_{pyr}$ (%); ±—analytical error.

The analysis of water-soluble organic carbon and nitrogen showed similar patterns with the content of total carbon and nitrogen. The maximum content was revealed for organic horizons. There was a gradual decrease in WSOC and WSON in mineral horizons. The organic horizons of the soils of the Komi Republic soils contain 1.74–13.00 mg g$^{-1}$ of water-soluble organic carbon, and the mineral horizons contain from 0.04 to 0.36$^{-1}$. The WSOC content in the organic horizons of the soils of the Krasnoyarsk region was 3.56–16.64 mg g$^{-1}$, and in mineral horizons 0.13–0.67 mg g$^{-1}$. The WSON content in soils is significantly lower. The organic horizons of the soils of the Komi Republic contain from 0.08 to 0.70 mg g$^{-1}$, and the mineral horizons contain 0.003–0.02 mg g$^{-1}$. The soils of the Krasnoyarsk region are characterized by similar values. The WSON content in the organic horizons was 0.18–0.87 mg g$^{-1}$, and in the mineral 0.002–0.02 mg g$^{-1}$.

Carbon ($C_{stock}$) and nitrogen ($N_{stock}$) stocks in the studied soils were calculated. There were no significant differences in the accumulation of both carbon and nitrogen in the studied soils. The calculation of the element stocks showed that the soils of the Komi Republic contain from 5.7 to 15.7 kg C m$^{-2}$ and from 0.47 to 0.92 kg N m$^{-2}$. The soils of the Krasnoyarsk region accumulate 6.9–12.5 kg C m$^{-2}$ and 0.61–0.83 kg N m$^{-2}$.

### 3.5. Densimetric Fractionation

That the predominance of the heavy fraction MaOM$_{>1.6}$ was characteristic for all the studied soils in the upper pyrogenic horizons (Table 4). The content of light densimetric fractions (fPOM$_{<1.6}$ and oPOM$_{<1.6}$) was lower. The content of heavy fractions in the soils of the Krasnoyarsk region varied from 94.4 to 97.8 mass%. The content of the fPOM$_{<1.6}$ was 0.2–2.9 mass%, and the oPOM$_{<1.6}$ varied from 0.6 to 1.8 mass%. Despite the low content, the main amount of carbon is contained in the light fractions. The carbon content in the fPOM$_{<1.6}$ fractions of the soils of the Krasnoyarsk region varied, 177–444 g kg$^{-1}$; the nitrogen content was 3.9–11.1 g kg$^{-1}$. The oPOM$_{<1.6}$ fractions contain 444–484 g kg$^{-1}$ of organic carbon and 5.5–9.4 g kg$^{-1}$ of nitrogen. The heavy fractions MaOM$_{>1.6}$ of the soils of the Krasnoyarsk region contain 7–31 g kg$^{-1}$ of carbon and 0.97–5.5 g kg$^{-1}$ of nitrogen.

The pyrogenic horizons of the soils of the Komi Republic contain from 92.8 to 98.7 mass% of heavy fractions. Their carbon content was 2.8–19 g kg$^{-1}$, and nitrogen content was 0.16–1.08 g kg$^{-1}$. This is significantly lower than in the light SOM fractions. The content of the fPOM$_{<1.6}$ fraction varied from 0.2 to 2.9 mass %, which is slightly higher than in the soils of Siberia. The light fractions of fPOM$_{<1.6}$ contain 300–425 g kg$^{-1}$ carbon and 3.9–10.1 g kg$^{-1}$ nitrogen. The second light fraction of oPOM$_{<1.6}$ accounts for 0.6–1.8 mass%. Its carbon content was 252–444 g kg$^{-1}$ and nitrogen 3.5–7.4 g kg$^{-1}$.

**Table 4.** Contribution of the fraction and carbon and nitrogen contents in the densimetric fractions of the $E_{pyr}$ horizon.

| Site | fPOM$_{<1.6}$ | | | | oPOM$_{<1.6}$ | | | | MaOM$_{>1.6}$ | | | |
|---|---|---|---|---|---|---|---|---|---|---|---|---|
| | Mass | C | N | C/N | Mass | C | N | C/N | Mass | C | N | C/N |
| | % | g kg$^{-1}$ | | | % | g kg$^{-1}$ | | | % | g kg$^{-1}$ | | |
| I-CS | 1.5 | 444 ± 16 | 11.1 ± 1.2 | 47 | 2.7 | 469 ± 16 | 9.4 ± 1.0 | 58 | 98.7 | 31 ± 5 | 1.8 ± 0.4 | 20 |
| II-CS | 0.2 | 177 ± 18 | 3.9 ± 0.8 | 53 | 0.9 | 404 ± 14 | 5.5 ± 1.1 | 86 | 92.8 | 7 ± 1.6 | 5.5 ± 1.1 | 1 |
| III-CS | 0.4 | 291 ± 29 | 6.7 ± 0.7 | 51 | 1.4 | 484 ± 17 | 8.8 ± 1.0 | 64 | 98.6 | 15 ± 4 | 0.97 ± 0.19 | 18 |
| I-EN | 0.2 | 330 ± 12 | 3.9 ± 0.8 | 99 | 0.6 | 444 ± 16 | 3.5 ± 0.7 | 148 | 94.4 | 2.8 ± 0.6 | 0.16 ± 0.05 | 20 |
| II-EN | 2.9 | 425 ± 15 | 10.1 ± 1.1 | 49 | 1.8 | 403 ± 14 | 7.4 ± 0.8 | 64 | 97.8 | 19 ± 4 | 1.08 ± 0.22 | 21 |
| III-EN | 1.3 | 300 ± 30 | 5.8 ± 1.2 | 60 | 1.5 | 252 ± 25 | 4.4 ± 0.9 | 67 | 97.0 | 4.6 ± 1 | 0.36 ± 0.11 | 15 |

Note: EN—European North (Komi Republic); CS—Central Siberia (Krasnoyarsk region); I, II, III—site numbers; fPOM$_{<1.6}$—free particulate organic matter; oPOM$_{<1.6}$—occluded particulate organic matter; MaOM$_{>1.6}$—mineral-associated organic matter; ±—analytical error.

### 3.6. Contents of PAHs

The total content of PAHs in the studied horizons ranges from 24 to 605 ng g$^{-1}$ (Table 5). The maximum values were found for the pyrogenic organic sub-horizons (311–605 ng g$^{-1}$), and the minimum values for the upper pyrogenic mineral horizons (24–169 ng g$^{-1}$). The PAH content in the organic horizons of the Komi Republic soils is 473–605 ng g$^{-1}$, and in the soils of the Krasnoyarsk region 311–393 ng g$^{-1}$. The upper mineral horizons of the soils of the Komi Republic contain 46–169 ng g$^{-1}$ PAHs, while 24–52 ng g$^{-1}$ is accumulated in similar soil horizons of the Krasnoyarsk region.

**Table 5.** PAH content in the upper horizons of the studied soils (ng g$^{-1}$).

| Site | Soil Hori-zon | Depth, cm | 2-Ring | | 3-Ring | | | 4-Ring | | | | 5-Ring | | | 6-Ring | | | Σ | ΣLP | ΣHP | C$_{PAHs}$ |
|---|---|---|---|---|---|---|---|---|---|---|---|---|---|---|---|---|---|---|---|---|---|
| | | | NP | ACE | FL | PHE | ANT | FLA | PYR | BaA | CHR | BbF | BkF | BaP | DahA | BghiP | IcdP | | | | |
| | | | | | | | | | | | Krasnoyarsk region | | | | | | | | | | |
| I-CS | O(e/a)$_{pyr}$ | 2–5 | 90 | 40 | 3 | 37 | 2 | 19 | 10 | 6 | 27 | 28 | 7 | 7 | 3 | 9 | 22 | 311 | 234 | 77 | 293 |
| | E$_{pyr}$ | 5–18 | 29 | 0 | 1 | 7 | 0 | 3 | 3 | 0 | 2 | 3 | 0 | 1 | 1 | 2 | 0 | 52 | 45 | 7 | 49 |
| II-CS | O(e/a)$_{pyr}$ | 2–5 | 55 | 8 | 3 | 56 | 2 | 29 | 14 | 7 | 23 | 44 | 8 | 9 | 3 | 28 | 49 | 337 | 296 | 141 | 319 |
| | E$_{pyr}$ | 5–10 | 15 | 0 | 1 | 6 | 0 | 2 | 1 | 1 | 1 | 2 | 0 | 0 | 0 | 0 | 0 | 30 | 28 | 2 | 28 |
| III-CS | O(e/a)$_{pyr}$ | 2–5 | 57 | 106 | 2 | 96 | 2 | 22 | 12 | 5 | 16 | 29 | 5 | 6 | 2 | 13 | 24 | 393 | 316 | 78 | 370 |
| | E$_{pyr}$ | 5–17 | 13 | 0 | 1 | 4 | 0 | 2 | 1 | 0 | 1 | 0 | 0 | 0 | 0 | 0 | 0 | 24 | 23 | 1 | 22 |
| | | | | | | | | | | | Komi Republic | | | | | | | | | | |
| I-EN | Oa$_{pyr}$ | 11–16 | 192 | 0 | 14 | 151 | 6 | 22 | 10 | 3 | 12 | 14 | 5 | 6 | 3 | 6 | 30 | 473 | 410 | 63 | 445 |
| | E$_{pyr}$ | 16–30 | 22 | 0 | 4 | 6 | 7 | 3 | 3 | 0 | 1 | 0 | 0 | 0 | 0 | 0 | 0 | 46 | 46 | 0 | 43 |
| II-EN | Oa$_{pyr}$ | 6–8 | 141 | 0 | 27 | 256 | 5 | 40 | 5 | 1 | 37 | 36 | 0 | 8 | 0 | 0 | 50 | 605 | 512 | 93 | 570 |
| | E$_{pyr}$ | 8–12 | 83 | 0 | 13 | 55 | 2 | 7 | 4 | 0 | 0 | 0 | 4 | 0 | 0 | 0 | 0 | 169 | 165 | 4 | 158 |
| III-EN | Oa | 4–6 | 160 | 0 | 14 | 140 | 6 | 43 | 24 | 7 | 16 | 14 | 6 | 10 | 0 | 4 | 30 | 476 | 411 | 65 | 448 |
| | E$_{pyr}$ | 6–10 | 25 | 0 | 10 | 45 | 1 | 4 | 4 | 0 | 1 | 0 | 0 | 1 | 0 | 0 | 0 | 90 | 89 | 1 | 85 |

Note: EN—European North (Komi Republic); CS—Central Siberia (Krasnoyarsk region); I, II, III—site numbers; NP—naphthalene; ACE—acenaphthene; FL—fluorene; PHE—phenanthrene; ANT—anthracene; FLA—fluoranthene; PYR—pyrene; BaA—benzo[a]anthracene; CHR—chrysene; BbF—benzo[b]fluoranthene; BkF—benzo[k]fluoranthene; BaP—benzo[a]pyrene; DahA—dibenzo[a,h]anthracene; BghiP—benzo[g,h,i]perylene; IcdP—indeno[1,2,3-c,d]pyrene; ΣLP—sum of light PAHs; ΣHP—sum of heavy PAHs; C$_{PAHs}$—recalculation taking into account the contribution of carbon in the individual molecules of the substances.

Among the isolated PAHs, light compounds (2-3-4 nuclear/ΣLP) predominate compared to heavy compounds (5 and 6 nuclear/ΣHP). The ΣLP content varies from 196 to 512 ng g$^{-1}$ in organic horizons or 58%–87% of the total PAH content in the horizon. In the mineral horizons of soils, ΣLP amount to 23–165 ng g$^{-1}$ or 86%–100% of the total PAH content. The content of ΣHP ranges from 63 to 141 ng g$^{-1}$ (13%–42%) in organic horizons and from 0 to 7 ng g$^{-1}$ (0%–14%) in mineral horizons. In addition, the data on the content of PAHs for carbon in the composition of PAHs was recalculated. It was found that the upper pyrogenic horizons contain from 93 to 96% carbon or 22–570 ng g$^{-1}$ in terms of carbon in the composition of PAHs.

## 4. Discussion

### 4.1. Morphological and Chemical Properties of Soils

Spruce forests grow on flat and slightly elevated areas, watersheds, and well-drained relief elements. Spruce forests form under more humid climatic conditions, and there is a high proportion of moisture-loving mosses and shrubs in the ground cover. Fires in spruce forests, in comparison to fires in pine forests, have more serious consequences. This is due to the development of surface root systems and the physiologically weak resistance of spruces to high temperatures.

Spruce forests predominate in the European north. In the Komi Republic, they occupy more than 50% of the forested area. The main spruce forests of the Komi Republic are concentrated in the subzones of the middle and northern taiga. The northern taiga is dominated by mature forests. Native spruce-green moss forests occupy about 35%. For the Krasnoyarsk region, spruce and fir forests occupy 17%. Native arrays of dark coniferous taiga have been preserved in small areas. Along the Yenisei river, spruce forests meet along the valleys of smaller rivers.

The studied soils of the native old-growth spruce forests of the Komi Republic and Krasnoyarsk region have similar morphological properties. Soils have a morphological structure typical of Retisols [41,42]. Soils are formed on loamy sediments and characterized by the migration of clay and iron compounds to the lower mineral horizons. As a result, a clay-poor and lighter upper E horizon is formed. In the lower part, a more intensely colored BE horizon is formed, which is richer in clay and iron compounds. Thus, the morphological features allow us to classify the studied soils as Glossic Stagnic Retisol (Siltic, Cutanic) [43].

It was found that well-defined coal inclusions occur in the upper horizons of the studied soils in the lower part of the organic and upper part of the mineral horizons, which is the result of old ground fires. Coal particles are an integral part of soils after a fire [44], which is consistent with our results and the data from the literature [45–47].

Our previously published studies have shown that carbonaceous particles can persist in the soil for up to 130 years [25,48]. In this case, it can be argued that pyrogenic morphological signs are diagnosed more than 200 years after the fire. Retisols probably have a high ability to store and accumulate pyrogenic carbon (PyC) in the upper horizons of soils. According to research [49], boreal forest soils store 1 Pg of carbon in the form of charcoal, which is equivalent to 1% of the total carbon stock of plants in boreal forests, and the average age of charcoal is 652 years. It is possible to make an assumption and attribute the studied soils to the pyrogenic subtype since the preservation of morphological pyrogenic signs was shown. The considered basic chemical properties of the studied spruce soils did not reveal a direct effect of pyrogenesis on them due to the long time that passed since the fire occurred. It is noted in the literature that the greatest changes in the chemical properties of soils occur in the first years after a fire [21,50,51].

Thus, it is shown that the studied Retisols of the Komi Republic and the Krasnoyarsk region are characterized by similar genetic, morphological, and chemical properties, despite the geographical remoteness of the regions. The studied morphological and physico-chemical properties of the soils of old native spruce forests can be used as standards for studying the influence of the pyrogenic factor on forest ecosystems and the ability of forests to recover after fires.

### 4.2. Soil Organic Matter

It was revealed that total water-soluble carbon (WSOC) consists of 99%–100% organic carbon. Ground cover plants are the main producers of WSOC and WSON [52,53]. It can be assumed that the ground vegetation cover of the studied areas recovered or is close to being restored to the characteristics of unburned forests, since more than 100 years passed since the last fire occurred at each of the sites. Therefore, the content of water-soluble carbon and nitrogen compounds is close to the values of unburned spruce forests and is comparable with the literature data on the spruce forests of the Komi Republic [21,48], larch forests of Central Siberia [20,25], and boreal forests of Finland [2]. In general, many authors note that

part of PyC can be transformed into WSOC [54]. However, the most noticeable changes in the content and composition of water-soluble carbon and nitrogen fractions are observed in the first years after fires.

The method of densimetric fractionation is less destructive than the procedure of chemical fractionation, and it is likely that the resulting fractions are directly related to the structure and functions of SOM in situ [38,39,55,56]. As a result of densimetric fractionation, the predominance of the heavy fraction ($MaOM_{>1.6}$) in the upper mineral horizons was revealed, which is consistent with the literature data [57–59]. It was found that the densimetric fractions differ significantly in the content of total organic carbon. The main carbon content is concentrated in the light fractions of SOM. The maximum concentrations are typical for the light fractions of $fPOM_{<1.6}$ and $oPOM_{<1.6}$, the minimum for the fraction of $MaOM_{>1.6}$. The statistical analysis showed that the carbon content correlates with the content of light SOM fractions ($r = 0.39–0.80$, $p < 0.05$). The total content of light fractions (the sum of light fractions) has a high Pearson coefficient with the content of carbon ($r = 0.62$, $p < 0.05$). The heavy fraction negatively correlates with the content of carbon ($r = –0.76$, $p < 0.05$). Perhaps this can be explained by the concentration of pyrogenic carbon (PyC) in these light physical fractions. PyC is considered as one of the most stable pools of carbon sequestration from the atmosphere [60]. According to the literature, pyrogenesis products are mainly part of light fractions [48,61]. Positive values of the correlation coefficient between light SOM fractions and the PAH content in the pyrogenic horizon $E_{pyr}$ ($r = 0.74$, $p < 0.05$) were revealed. The most significant correlation is characteristic of the $fPOM_{<1.6}$ fraction ($r = 0.93$, $p < 0.05$). The correlation coefficients of $fPOM_{<1.6}$ with individual PAH compounds are: naphthalene ($r = 0.93$, $p < 0.05$), phenanthrene ($r = 0.81$, $p < 0.05$), fluorine ($r = 0.88$, $p < 0.05$), fluoranthene ($r = 0.77$, $p < 0.05$), pyrene ($r = 0.81$, $p < 0.05$), and benzo[k]fluoranthene ($r = 0.86$, $p < 0.05$). However, this requires additional studies of the composition and properties of densimetric fractions. It is necessary to use modern analytical methods, such as CP-MAS $^{13}$C-NMR spectroscopy, the content of PAHs and benzene polycarboxylic acids (BPCA), to search for signs of PyC pyrogenic carbon directly in fractions.

Bulk density is one of the most important physical soil properties [62], which plays a key role in calculating carbon and nitrogen reserves in soils. A number of studies indicate an increase in the density of organic horizons after fires due to saturation with pyrogenesis products (soot and coal) [23,63]. Thus, it can be assumed that the high density values of the soils of the Komi Republic are caused by the participation of pyrogenic components, which leads to an increase in the reserves of elements in organic horizons. It is likely that the forests of the Komi Republic were exposed to more frequent and more intense fires than the forests of the Krasnoyarsk region.

Organic horizons of boreal forest soils represent a permanent cover of fallen foliage and wood debris [64]. They are poorly susceptible to decomposition due to low temperatures and high humidity; therefore, they are well preserved and play a significant role as a nutrient pool. In addition, the organic horizons of soils after fires are denser and are represented by pyrogenesis products, layers of coals mixed with fresh vegetation. The contribution of mineral and organic horizons to the reserves of elements is shown separately. The contribution of organic horizons to carbon stocks varies from 14 to 52%. Most of the carbon and nitrogen reserves are concentrated in mineral horizons (48%–86%). An important indicator is the contribution of the upper mineral pyrogenic horizon ($E_{pyr}$) to the total reserves, which is a significant part, from 7 to 45%, which in some cases is comparable to the contribution of organic horizons. Epyr pyrogenic horizons contain 0.5–5.5 kg m$^{-2}$ of carbon and 0.05–0.26 kg m$^{-2}$ of nitrogen.

The results obtained generally coincide with previously published data for the soils of the spruce forests of the European Northeast [65–67], and in general for the soils of the Komi Republic [68] and Central Siberia [25,69,70]. According to a recent assessment of carbon and nitrogen reserves in the predominant soils of the Komi Republic, reserves range from 5.17 to 73.23 kg m$^{-2}$ [68]. However, for the first time, the contribution of pyrogenic

mineral horizons to the total carbon and nitrogen reserves of the spruce forests soils of the Komi Republic and the Krasnoyarsk region was determined.

*4.3. PAH Content*

It is known that forest fires are the main natural source of polyaromatic hydrocarbons [71–73]. Organic and mineral horizons differ both in the total content of PAHs and in individual compounds. The low content of polyarenes in the mineral horizons of soils is largely due to the weak ability of PAHs to migrate and their low solubility. It was revealed that light PAHs predominate in the studied soils, and their proportion varies from 58% in organic horizons to 100% in upper mineral horizons. The predominant compounds are naphthalene (NP) and phenanthrene (PHE), and their share of the total amount of PAHs is 12%–41%. The remaining compounds are in significantly lower amounts. Heavy PAHs characterized by lower solubility [74] were absent or were in minimal concentrations in mineral horizons.

According to [75], fluorene, naphthalene, and pyrene are among the predominant PAHs in soils not subject to fires. Our plots were burned 100–200 years ago, so the content of naphthalene in them can probably be considered close to the values of unburned forests and considered standard. The presence of phenanthrene and heavier structures (including benzo[b]fluoranthene) is associated with their formation from the wax components of coniferous trees during fires. It was also shown in [76] that, when burning coniferous trees (spruce and pine), the concentration of PAHs formed is maximal.

There is an increased PAH content in the soils of the Komi Republic compared to the soils of the Krasnoyarsk region by 1.5–3 times. An increase in the total content of PAHs in pyrogenic soil horizons due to 2–3 nuclear PAHs is evidence of the influence of fire. The main difference is due to the content of naphthalene, whose molecules have a high aromaticity. The concentration of naphthalene increases mainly in high-intensity fires. It is noted in the literature [21,72] that the formation of low molecular weight PAHs depends more on fire conditions, and the content of heavy PAHs is mainly related to the characteristics of biomass and the species diversity of vegetation. It can be assumed that, at a higher fire intensity, a greater amount of polyarenes will be formed and the qualitative composition of PAHs in soil horizons will be different during fires of different intensity. However, at the same time, it is difficult to use PAHs as indicators of the intensity of fires due to a large number of other impact factors. When studying fresh post-fire sites, both light and heavy polyarenes can be used as indicators. For old areas after a fire, it is better to use heavy structures, as they are less prone to decomposition. Therefore, differences in the content of naphthalene in the soils of the studied regions can be associated both with different intensity of fires and with the composition of vegetation. In this work, we did not set ourselves the task of studying the effect of fire intensity on the composition of PAHs in soils, but this issue may become a topic for our further research.

We calculated several diagnostic coefficients (Appendix A) that allowed us to determine the origin of PAHs in soils [77–82]. According to the calculated ratios of BaA/228 and BaA/(BaA+CHR), the petrogenic origin of PAHs in these soils was shown. The ANT/(ANT+PHE) and PHE/ANT ratios indicate the effect of pyrogenesis on the formation of PAHs only in the upper mineral horizon of the soil of the I-EN site. For the remaining samples, the coefficients demonstrate the formation of PAHs from rocks. Pyrogenic origin of PAHs in some horizons of the studied soils showed the following relations (PYR+FLA)/(CHR+PHE) and (PYR+BaP)/CHR+PHE). The ratio (PYR+BaP)/CHR+PHE) revealed that the PAHs contained in the soils of the Krasnoyarsk region are of pyrogenic origin. For the soils of the Komi Republic, according to these ratios, PAHs were formed as a result of combustion only in the upper mineral soil horizon of the I-EN site and the organic horizon of the III-EN site.

The coefficient FLA/(FLA+PYR) was the most useful to diagnose the pyrogenic origin of PAHs (Figure 3, Table A2).

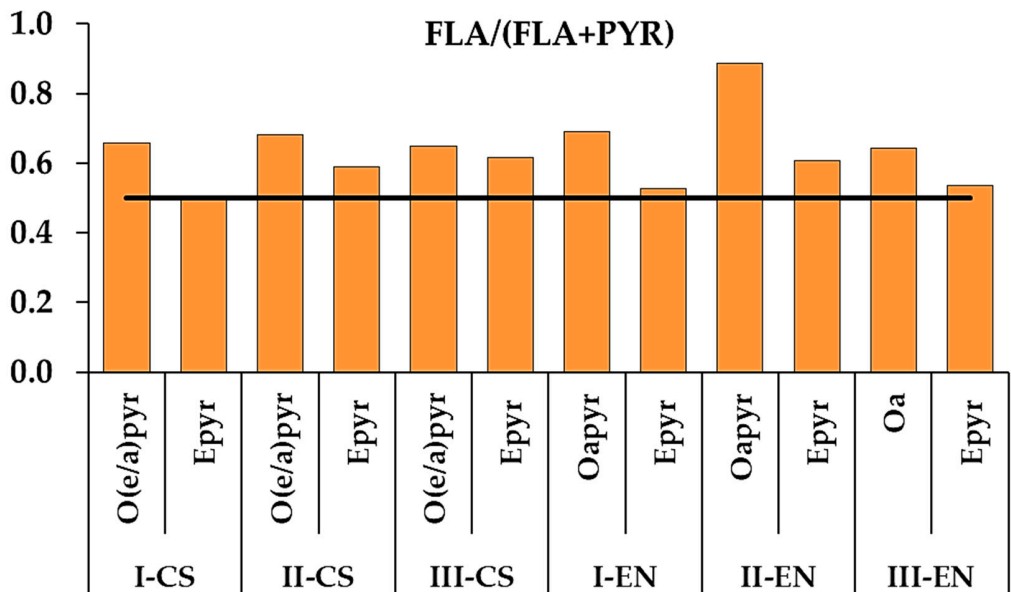

**Figure 3.** FLA/(FLA+PYR) ratio in the investigated pyrogenic horizons. The line marks the boundary after which (>0.5) the ratio indicates the pyrogenic origin of PAHs. EN—European North (Komi Republic); CS—Central Siberia (Krasnoyarsk region); I, II, III—site numbers.

According to the literature data [83–85], it reflects the nature of the burning of wood, herbs, and other plant residues. According to this ratio, all investigated horizons have pyrogenic origin of PAHs, which is also proved by dendrochronological studies at these sites. This is especially evident for organic horizons, where the coefficient FLA/(FLA+PYR) indicates the dominant influence of biomass combustion during the formation of PAHs.

Thus, it can be argued that pyrogenic signs in the soils of spruce forests after 100–200 years are reflected not only in morphological changes in soils (the presence of coals, soot, etc.), but also in an increase in PAH content, which for a long time remains a criterion for determining pyrogenic changes in soils after fires.

## 5. Conclusions

The study of the soils of indigenous spruce forests of the Komi Republic and the Krasnoyarsk Territory made it possible to reveal the patterns of the influence of pyrogenesis and the preservation of pyrogenic characteristics in soils after a fire. All the studied soils are characterized by the fact that more than a century has passed since the last fires. However, pyrogen products are preserved in soils, diagnosed by the morphological features of soils and the specifics of soil organic matter. The studied soils are characterized by similar morphological and chemical properties. It was revealed that, after fires, organic horizons are compacted, and morphologically diagnosed carbonaceous inclusions are preserved in organic $Q_{pyr}$ and mineral $E_{pyr}$ horizons. Significant differences in the stocks of carbon and nitrogen, and the content of water-soluble organic matter between the soils of the studied regions were not revealed. However, an increase in the contribution of the $E_{pyr}$ pyrogenic mineral horizon to the total stocks of carbon and nitrogen were established. It is suggested that light densimetric fractions are an important pool for preserving pyrogenic carbon, which requires additional research.

A high content of PAHs in pyrogenic organic and upper mineral horizons was found 100–200 years after the fire. The calculated coefficients FLA/(FLA+PYR) confirm the pyrogenic origin of the isolated PAHs in the studied soils. It is shown that the content of PAHs can be used as markers of pyrogenesis. The obtained materials can be the basis for further monitoring observations in the study regions (European North and Central Siberia).

**Author Contributions:** Conceptualization, V.V.S. and A.A.D.; Investigation, V.V.S., A.A.D. and I.N.K.; Methodology, V.V.S., A.A.D. and E.V.Y.; Project administration, V.V.S. and A.A.D.; Writing—original draft, V.V.S.; Writing—review and editing, V.V.S., A.A.D., E.V.Y. and I.N.K. All authors have read and agreed to the published version of the manuscript.

**Funding:** This work was supported by the Russian Foundation for Basic Research (RFBR) under Grant No. 19-29-05111 mk.

**Institutional Review Board Statement:** Not applicable.

**Informed Consent Statement:** Not applicable.

**Data Availability Statement:** The data presented in this study are available upon request from the corresponding author.

**Acknowledgments:** The authors express their gratitude to A.S. Prokushkin for his help in organizing work on the territory of the Krasnoyarsk region.

**Conflicts of Interest:** The authors declare that they have no known competing financial interests or personal relationships that could have appeared to influence the work reported in this paper.

## Appendix A

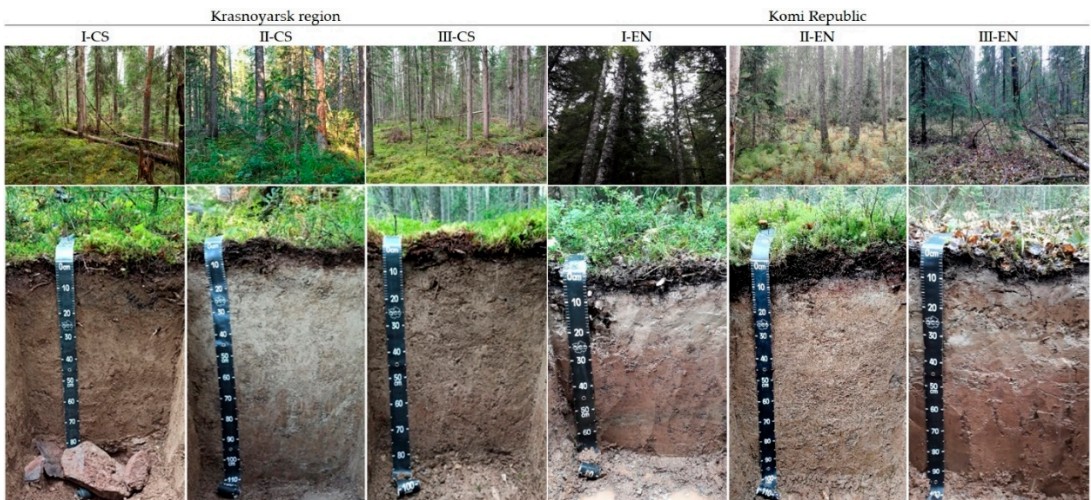

**Figure A1.** Photos of forests and soil profiles. EN—European North (Komi Republic); CS—Central Siberia (Krasnoyarsk region); I, II, III—site numbers.

**Table A1.** Texture of the studied soils.

| Site | Soil Horizon | Depth, cm | Sand | | | | | Silt | | Clay | Name According to the Ferre Triangle |
|---|---|---|---|---|---|---|---|---|---|---|---|
| | | | 2.000–1.000 | 1.000–0.500 | 0.500–0.250 | 0.250–0.100 | 0.100–0.050 | 0.050–0.020 | 0.020–0.002 | <0.002 | |
| | | | Very Corse | Corse | Medium | Fine | Very Fine | Coarse | Medium and Fine | | |
| | | | | | | Krasnoyarsk region | | | | | |
| I-CS | $E_{pyr}$ | 5–18 | 0 | 1 | 1 | 2 | 4 | 30 | 43 | 19 | Silt loam |
| | E | 18–30 | 0 | 1 | 1 | 2 | 5 | 26 | 44 | 21 | |
| | BE | 30–55 | 0 | 1 | 1 | 2 | 4 | 33 | 41 | 19 | |
| | Bt | 55–75 | 0 | 2 | 1 | 3 | 3 | 25 | 41 | 25 | |
| | BC | 75–90 | 1 | 8 | 7 | 13 | 5 | 18 | 27 | 22 | Loam |

Table A1. *Cont.*

| Site | Soil Horizon | Depth, cm | Sand | | | | | Silt | | Clay | Name According to the Ferre Triangle |
|---|---|---|---|---|---|---|---|---|---|---|---|
| | | | 2.000–1.000 | 1.000–0.500 | 0.500–0.250 | 0.250–0.100 | 0.100–0.050 | 0.050–0.020 | 0.020–0.002 | <0.002 | |
| | | | Very Corse | Corse | Medium | Fine | Very Fine | Coarse | Medium and Fine | | |
| | | | | | | Krasnoyarsk region | | | | | |
| II-CS | $E_{pyr}$ | 5–10 | 1 | 1 | 1 | 2 | 6 | 2 | 61 | 27 | |
| | E | 10–35 | 1 | 2 | 1 | 2 | 6 | 15 | 51 | 23 | Silt loam |
| | E2 | 35–60 | 0 | 1 | 1 | 2 | 5 | 26 | 45 | 20 | |
| | BE | 60–75 | 0 | 1 | 1 | 1 | 4 | 23 | 36 | 35 | Silty clay loam |
| | Bt | 75–90 | 0 | 0 | 0 | 1 | 2 | 8 | 40 | 49 | Silty clay |
| | BC | 90–110 | 0 | 0 | 0 | 1 | 2 | 15 | 31 | 51 | |
| III-CS | $E_{pyr}$ | 5–17 | 0 | 1 | 1 | 2 | 4 | 31 | 41 | 21 | |
| | E | 17–30 | 0 | 1 | 1 | 2 | 3 | 31 | 41 | 21 | Silt loam |
| | E2 | 30–50 | 0 | 1 | 1 | 2 | 4 | 21 | 48 | 23 | |
| | BE | 50–70 | 0 | 2 | 1 | 2 | 2 | 26 | 29 | 38 | Silty clay loam |
| | Bt | 70–90 | 0 | 1 | 1 | 3 | 3 | 29 | 24 | 40 | |
| | | | | | | Komi Republic | | | | | |
| I-EN | $E_{pyr}$ | 16–30 | 1 | 3 | 14 | 17 | 2 | 8 | 23 | 32 | Clay loam |
| | E2 | 30–50 | 0 | 3 | 12 | 16 | 2 | 9 | 24 | 33 | |
| | BE | 50–75 | 0 | 2 | 11 | 16 | 2 | 12 | 23 | 34 | |
| | Bt | 75–90 | 0 | 1 | 6 | 8 | 1 | 18 | 27 | 39 | Silty clay loam |
| II-EN | $E_{pyr}$ | 8–12 | 0 | 0 | 0 | 1 | 2 | 12 | 32 | 53 | |
| | E | 12–20 | 0 | 0 | 0 | 1 | 2 | 13 | 31 | 52 | |
| | BE | 20–35 | 0 | 0 | 0 | 0 | 1 | 15 | 32 | 50 | Silty clay |
| | BE2 | 35–55 | 0 | 0 | 0 | 1 | 1 | 15 | 34 | 49 | |
| | Bt | 55–80 | 0 | 0 | 0 | 1 | 2 | 16 | 33 | 48 | |
| | BC | 80–100 | 0 | 0 | 0 | 0 | 2 | 15 | 29 | 53 | |
| III-EN | $E_{pyr}$ | 6–10 | 0 | 0 | 1 | 2 | 9 | 6 | 20 | 62 | |
| | E | 10–25 | 1 | 1 | 1 | 3 | 8 | 7 | 20 | 60 | Clay |
| | E2 | 25–35 | 0 | 0 | 0 | 1 | 6 | 6 | 20 | 65 | |
| | BE | 35–45 | 0 | 0 | 1 | 2 | 5 | 14 | 23 | 56 | |
| | Bt | 50–70 | 0 | 0 | 0 | 1 | 4 | 19 | 26 | 50 | Silty clay |
| | BC | 70–100 | 0 | 0 | 1 | 1 | 6 | 18 | 24 | 50 | |

Note: EN—European North (Komi Republic); CS—Central Siberia (Krasnoyarsk region); I, II, III—site numbers.

Table A2. Diagnostic ratios of individual PAHs in soils.

| Site | Horizon | Depth, cm | (PYR+FLA)/ (CHR+PHE) | (PYR+BaP)/ CHR+PHE) | ANT/(ANT+PHE) | PHE/ ANT | FLA /(FLA+PYR) | BaA/228 | BaA/ (BaA+CHR) |
|---|---|---|---|---|---|---|---|---|---|
| Pyrogenic Origin Index | | | >0.5 | >0.1 | >0.1 | <10 | >0.5 | >0.5 | >0.35 |
| | | | | | Krasnoyarsk region | | | | |
| I-CS | $O(e/a)_{pyr}$ | 2–5 | 0.45 | 0.27 | 0.04 | 23.38 | 0.66 | 0.03 | 0.17 |
| | $E_{pyr}$ | 5–18 | 0.56 | 0.35 | 0.03 | 29.00 | 0.50 | 0.00 | 0.17 |
| II-CS | $O(e/a)_{pyr}$ | 2–5 | 0.55 | 0.30 | 0.03 | 32.00 | 0.68 | 0.03 | 0.23 |
| | $E_{pyr}$ | 5–10 | 0.48 | 0.20 | 0.03 | 30.00 | 0.59 | 0.00 | 0.31 |
| III-CS | $O(e/a)_{pyr}$ | 2–5 | 0.30 | 0.16 | 0.02 | 63.67 | 0.65 | 0.02 | 0.22 |
| | $E_{pyr}$ | 5–17 | 0.53 | 0.27 | 0.05 | 18.00 | 0.62 | 0.00 | 0.24 |
| | | | | | Komi Republic | | | | |
| I-EN | $Oa_{pyr}$ | 11–16 | 0.20 | 0.10 | 0.04 | 26.56 | 0.69 | 0.01 | 0.21 |
| | $E_{pyr}$ | 16–30 | 0.78 | 0.37 | 0.53 | 0.88 | 0.53 | 0.00 | 0.00 |
| II-EN | $Oa_{pyr}$ | 6–8 | 0.15 | 0.04 | 0.02 | 50.29 | 0.89 | 0.00 | 0.02 |
| | $E_{pyr}$ | 8–12 | 0.20 | 0.08 | 0.04 | 22.80 | 0.61 | 0.00 | 0.00 |
| III-EN | Oa | 4–6 | 0.43 | 0.22 | 0.04 | 24.43 | 0.64 | 0.03 | 0.32 |
| | $E_{pyr}$ | 6–10 | 0.18 | 0.09 | 0.02 | 51.00 | 0.54 | 0.00 | 0.00 |

Note: EN—European North (Komi Republic); CS—Central Siberia (Krasnoyarsk region); I, II, III—site numbers; PHE—phenanthrene; ANT—anthracene; FLA—fluoranthene; PYR—pyrene; BaA—benzo[a]anthracene; CHR—chrysene; BaP—benzo[a]pyrene.

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
