# Peer review of "Fire Impact on Carbon Pools and Basic Properties of Retisols in Native Spruce Forests of the European North and Central Siberia of Russia"

_forests, doi:10.3390/f13071135_

Round 1

Reviewer 1 Report

Manuscript addresses the general topic of fires, their impact on ecosystems, and is the basis for future research. The Authors present new possibilities for monitoring ecosystems that have been exposed to fires. However, there was some confusion in the manuscript.

Lines 65-68: it would be worth summarizing the aims of the research to formulate the hypothesis / hypotheses

Lines 73-75: a map or coordinates of research plots would be welcome

Line 85: Table1  Please provide an explanation of the study site abbreviations

part of the manuscript - Results The Authors often compare the results, but this is not supported by statistical analysis.

Line 180: please provide an explanation of the horizons and sites abbreviations, this would make the table easier to read. The same in the case of Figure 1 - line 212

Lines 242 (Table 3), 270 (Table 4), 281 (Table 5)  please provide an explanation of the abbreviations to horizons and sites

Reviewer 2 Report

Forests-1754640-V2

Fire impact on the carbon pools and basic properties of Retisols in native spruce forests of European North and Central Siberia of Russia

Startsev et al. present a description of forest soils at several sites in two regions in Russia, under predominantly spruce forests that are suggested to have burned between 100 and 200 years ago. The time since fire at each site is stated precisely (e.g. 146 years at the first site in the Krasnoyarsk region), but the method by which these dates were determined is not described. No previous studies are clearly cited to support these times-since-fires, though in the Discussion section there is a brief mention of "dendrochonological studies" but no specific citations.

While this study is largely descriptive and does not attempt to deeply analyse the observed patterns in soil carbon and other properties, and is presented as largely a baseline dataset to support future studies, there are some analytical arguments throughout. The time since fire at each site is a crucial part of these analyses; the descriptions of soil properties are linked to these ages at every step. Yet we do not know how these ages were determined. Furthermore, fire intensity is argued to be another important factor, especially in regards to the composition of fire-created organic compounds and soil features such as coal particles and soot. Again, citations or methodological descriptions are missing that would clarify the relationships between soil organic matter composition and forest fires. Potentially confounding variables – such as the species composition of forests separated by hundreds of kilometres – are mentioned in passing but given little consideration.

My detailed comments, below, may mostly be addressed by a good description in the Materials and Methods section of how time since fire was determined, and an estimate of the precision of these times. Further description of the relationship between fire intensity and carbon compound composition, especially naphthalene, is also necessary to support arguments made in the Discussion section.

2. Materials and Methods

How was the time since a fire in a forest site determined? Are there reliable records of fires dating back 200 years in these wilderness areas? Have previous researchers in these areas determined the dates of fires?

4. Discussion

LN 330 – how was the age of charcoal determined? If the average of 652 years came from another study, please provide a citation. If it was calculated in this study, please describe the dating procedure in the Methods section.

LN 374-376 – if you have data on time since fire, do you also have estimates of fire intensity, independent of the composition of PAHs and other products of fires? Can you use the composition of PAHs found in the soil layers at your sites to estimate fire intensity?

LN 410-414 – "is considered" and "it is obvious" are not substitutes for citations of previous, rigorous studies to support these arguments. Why would naphthalene increase in higher intensity fires? Are there other ways to increase naphthalene – previous sentences mention differences in biomass and species diversity; do some plants contribute more naphthalene than others when burned?

LN 416 – "passed by fires 100-150 years ago [20]." Reference [20] describes forest soils that experienced fires from 2 to 16 years prior to their investigations – how does this reference support the stated date range since fire for your sites in the Komi Republic?

LN 436 – which dendrochonological studies support your assertion of the time since fire in your study sites?

Typos and other small errors

LN 103 – "bilk density" should be "bulk density"

LN 442 – "soot" is repeated
